# A comprehensive genome-wide cross-trait analysis of sexual factors and uterine leiomyoma

Xueyao Wu[1,2☯], Changfeng Xiao[1☯], Danielle Rasooly[3], Xunying Zhao[1], Cynthia Casson Morton[4,5,6,7], Xia Jiang[1,8,9]*, C. Scott Gallagher[10]*

1 Department of Epidemiology and Biostatistics, West China School of Public Health and West China Fourth Hospital, Sichuan University, Chengdu, Sichuan, China, 2 National Cancer Institute, Rockville, Maryland, United States of America, 3 Division of Aging, Department of Medicine, Brigham and Women's Hospital, Harvard Medical School, Boston, Massachusetts, United States of America, 4 Department of Obstetrics and Gynecology, Brigham and Women's Hospital, Harvard Medical School, Boston, Massachusetts, United States of America, 5 Department of Pathology, Brigham and Women's Hospital, Harvard Medical School, Boston, Massachusetts, United States of America, 6 Broad Institute of MIT and Harvard, Cambridge, Massachusetts, United States of America, 7 Manchester Centre for Audiology and Deafness, Manchester Academic Health Science Center, University of Manchester, Manchester, United Kingdom, 8 Department of Clinical Neuroscience, Center for Molecular Medicine, Karolinska Institutet, Solna, Stockholm, Sweden, 9 Department of Nutrition and Food Hygiene, West China School of Public Health and West China Fourth Hospital, Sichuan University, Chengdu, China, 10 Department of Genetics, Harvard Medical School, Boston, Massachusetts, United States of America

☯ These authors contributed equally to this work.
* xia.jiang@ki.se (XJ); cscottgallagher@gmail.com (CSG)

**Data Availability Statement:** The data underlying the results presented in the study are available to researchers upon application to UK Biobank (https://www.ukbiobank.ac.uk/) or are (partially)

## Abstract

Age at first sexual intercourse (AFS) and lifetime number of sexual partners (NSP) may influence the pathogenesis of uterine leiomyoma (UL) through their associations with hormonal concentrations and uterine infections. Leveraging summary statistics from large-scale genome-wide association studies conducted in European ancestry for each trait ($N_{AFS}$ = 214,547; $N_{NSP}$ = 370,711; $N_{UL}$ = 302,979), we observed a significant negative genomic correlation for UL with AFS ($r_g$ = −0.11, $P$ = 7.83×10⁻⁴), but not with NSP ($r_g$ = 0.01, $P$ = 0.62). Four specific genomic regions were identified as contributing significant local genetic correlations to AFS and UL, including one genomic region further identified for NSP and UL. Partitioning SNP-heritability with cell-type-specific annotations, a close clustering of UL with both AFS and NSP was identified in immune and blood-related components. Cross-trait meta-analysis revealed 15 loci shared between AFS/NSP and UL, including 7 novel SNPs. Univariable two-sample Mendelian randomization (MR) analysis suggested no evidence for a causal association between genetically predicted AFS/NSP and risk of UL, nor vice versa. Multivariable MR adjusting for age at menarche or/and age at natural menopause revealed a significant causal effect of genetically predicted higher AFS on a lower risk of UL. Such effect attenuated to null when age at first birth was further included. Utilizing participant-level data from the UK Biobank, one-sample MR based on genetic risk scores yielded consistent null findings among both pre-menopausal and post-menopausal females. From a genetic perspective, our study demonstrates an intrinsic link underlying sexual factors (AFS and NSP) and UL, highlighting shared biological mechanisms rather than direct causal

publicly available summary statistics. GWAS summary statistics for age at first sexual intercourse, lifetime number of sexual partners, and uterine leiomyoma (without 23andMe) are publicly available. To request access to 23andMe GWAS summary statistics, please visit https://research.23andme.com/dataset-access/.

**Funding:** Funding to C.C.M. was provided by the U. S. National Institute of Health (NIH)/Eunice Kennedy Shriver National Institute of Child Health and Human Development (NICHD, https://www.nichd.nih.gov/) (No. HD060530). The funders had no role in study design, data collection and analysis, decision to publish, or preparation of the manuscript.

**Competing interests:** The authors have declared that no competing interests exist.

effects. Future studies are needed to elucidate the specific mechanisms involved in the shared genetic influences and their potential impact on UL development.

## Author summary

While uterine leiomyoma (UL) is associated with multiple female reproductive characteristics, little is known regarding its relationship with sexual factors including age at first sexual intercourse (AFS) and lifetime number of sexual partners (NSP). In this study, leveraging large-scale summary- and participant-level genetic variation data, we performed a comprehensive genome-wide cross-trait analysis to assess the shared genetic basis, pleiotropic effects, and bidirectional causal relationships between AFS, NSP, and UL. Our findings provide substantial evidence for a considerable overlapping biology, highlighted by significant genetic correlations, clustering of cell-type-specific annotations, and the presence of multiple shared loci across traits. Nevertheless, we did not identify any compelling evidence supporting a direct causal effect of AFS or NSP on UL risk, nor did we observe a plausible causal role of UL in influencing sexual factors. Collectively, our study underscores that the observed shared heritabilities between sexual factors and UL are primarily driven by common biological mechanisms rather than a causal relationship.

## Introduction

As a major source of morbidity for females of reproductive age, uterine leiomyoma (UL) has been associated with multiple female reproductive characteristics including parity, age at first or last childbirth, and age at menarche (AAM)[1,2]. In contrast, sexual factors such as age at first sexual intercourse (AFS) and lifetime number of sexual partners (NSP) receive little empirical attention, even with a decrease in AFS and an increase in NSP being observed in many contemporary societies [3–5]. In addition to impacting ovulation and hormonal concentrations [6,7], sexual factors may expose participants to infectious agents in the uterus [8,9]. Therefore, low AFS or high NSP may correlate with alterations in sex hormones and/or an increased susceptibility to uterine infections, both of which have previously been hypothesized to play an etiologic role in UL pathogenesis [10–14]. Leveraging data from 181 UL-affected females and 133 healthy referents, Pakiz et al. (2020) found a significantly decreased risk of multiple UL associated with higher AFS (OR = 0.27, 95% CI = 0.14–0.52, 15]. However, results from retrospective studies are susceptible to bias, including confounding, and reverse causality; and no prospective study has been conducted to date to investigate the phenotypic associations between AFS or NSP with UL.

In the absence of well-designed observational studies, linking phenotypes through genetics may help uncover the intrinsic mechanisms underlying complex traits. Indeed, genetic regulation of UL and sexual factors have been highlighted by discoveries from recent large-scale genome-wide association studies (GWASs). The SNP (single nucleotide polymorphism)-heritabilities of UL, AFS, and NSP have been quantified as 3–13% [CI = 2–22%] [16,17], 12~23% [CI = 7–28%] [18], and 13% [CI = 12–13%] [19], suggesting a non-trivial genetic background for each trait. Further, several common loci have been identified, harboring genes related to follicle stimulating hormone (*FSHB*), implantation (*ESR1*), and embryogenesis (*WNT2*, 16,18,19]. These results suggest that sexual factors and UL might be driven by common biology, though the extent and nature of such links remain unclear. Examining the precise roles of

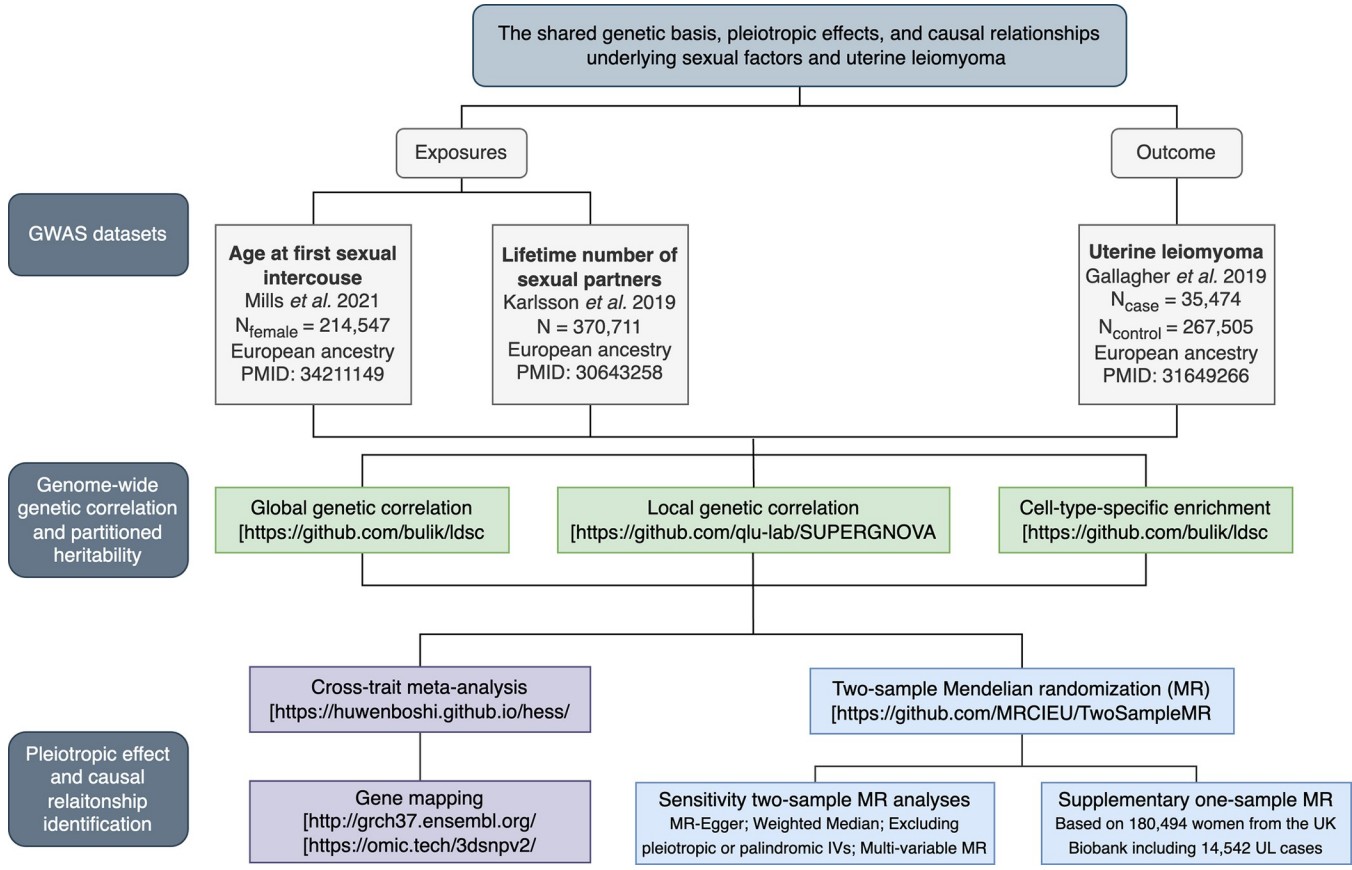

**Fig 1. Overall study design of genome-wide cross-trait analysis.** GWAS summary statistics for each trait of interest were retrieved from publicly available GWAS(s). A global genetic correlation analysis between sexual factors and uterine leiomyoma was conducted. The estimated global genetic correlation was further dissected at LD-defined regions. Functional annotation analysis within 396 cell-type-specific annotations was conducted on three traits by using genome-wide SNPs. Cross-trait meta-analysis was applied to identify pleiotropic loci, and a comprehensive Mendelian randomization analysis was used to infer putative causal relationships.

AFS and NSP in UL pathogenesis may advance our understanding of the etiology of disease and possibly aid in the prevention of UL.

Recent advances in statistical genetics have yielded a range of methods, enabling comprehensive genome-wide cross-trait analyses to characterize extensively the shared genetic architecture and causal link across traits [20]. Here, we applied these methods to perform a large-scale genome-wide cross-trait analysis between sexual factors and UL. Specifically, leveraging large-scale genetic variation data at both the summary and participant levels, we quantified the shared genetic basis, detected pleiotropic effects, and assessed putative bidirectional causal relationships underlying AFS, NSP, and UL. The overall study design is shown in **Fig 1**.

## Materials and methods

### Ethics statement

The GWAS summary statistics used in the present study are aggregated level of data which do not contain any personal information. The original GWAS (or participating cohorts) have obtained ethical approval from relevant ethics review committees (e.g., the Department of Sociology, University of Oxford, the Partners HealthCare System Human Research Committee [Women's Health Study/Women's Genome Health Study], the Ethical Committee of the

Northern Ostrobothnia Hospital District [Northern Finnish Birth Cohort], the Human Research Ethics Committee at the QIMR Berghofer Medical Research Institute and the Australian Twin Registry [QIMR Berghofer Medical Research Institute], the North West Multi-centre Research Ethics Committee [UK Biobank], Ethical and Independent Review Services [an external institutional review board; 23andMe], and the Institutional Review Boards at Harvard T.H. Chan School of Public Health and Brigham and Women's Hospital [Partners Human Research Committee] [Nurses' Health Study II]). UK Biobank received ethical approval from the Research Ethics Committee (reference 11/NW/0382), and all participants provided written informed consent.

## GWAS summary data

**Sexual factors.** For AFS, we obtained association results from the UK Biobank for 214,547 females of European ancestry [18]. For NSP, due to the unavailability of formally published sex-specific summary statistics, we obtained association results from the UK Biobank for 370,711 females and males of European ancestry [19]. AFS and NSP in the UK Biobank were assessed through the questions, "What was your age when you first had sexual intercourse? (sexual intercourse includes vaginal, oral, or anal intercourse)", and "About how many sexual partners have you had in your lifetime?", respectively. Both GWAS controlled for the top ten genetic principal components, birth year, and sex (in the sex-combined analysis for NSP). Since the GWAS for AFS identified genetic associations unique to each sex (e.g., 2 female-specific lead SNPs which had no effect at all in males), we applied clumping within a 500kb window ($r^2 > 0.02$) using female-specific summary data and identified 113 independent index SNPs ($P < 5 \times 10^{-8}$). These index SNPs were used as candidate instrumental variables (IVs). For NSP, candidate IVs were determined as the 117 trait-associated ($P < 5 \times 10^{-8}$) index SNPs reported by the GWAS. Details on characteristics of IVs are shown in **S1 and S2 Tables**. We also retrieved full set summary statistics of both AFS and NSP for additional analysis.

**Uterine Leiomyoma.** For UL, we obtained association results from the hitherto largest GWAS meta-analysis including data from five participating cohorts of the FibroGENE consortium (Women's Genome Health Study [WGHS], Northern Finnish Birth Cohort [NFBC], QIMR Berghofer Medical Research Institute [QIMR], UK Biobank, and 23andMe), comprising 302,979 female individuals (35,474 cases and 267,505 controls) of European ancestry [16]. UL was determined based on either self-report (via a single question, such as "Have you ever been diagnosed with uterine fibroids?") or clinical documentation (e.g., hospital-linked medical records). For cohorts in which information regarding previous hysterectomy was collected, females with a history of hysterectomy were excluded. Age, genetic relatedness, and body mass index were adjusted for in the association analyses. Top-associated SNPs in the meta-analysis reaching genome-wide significance ($P < 5 \times 10^{-8}$) were reported. We extracted relevant information for the 27 UL-associated autosomal SNPs and used those SNPs as IVs. Details on characteristics of the UL-associated IVs are shown in **S3 Table**. We also retrieved full summary statistics for UL GWAS (with the inclusion of data from 23andMe).

A table summarizing the information of each included GWAS is shown as **S4 Table**. For all analyses, the human reference genome build 37 (hg19) was used.

## Statistical analysis

**Global genetic correlation analysis.** We applied linkage-disequilibrium score regression (LDSC) [21] to estimate genome-wide genetic correlation aggregating effects of all SNPs, including those that do not reach genome-wide significance. LDSC quantifies the average sharing of genetic effect between two traits with the merits of using only GWAS summary

statistics and is not biased by sample overlap. The estimate ranges between −1 and +1, with +1 indicating a perfect positive correlation and −1 indicating a perfect negative correlation. A Bonferroni-corrected *P*-value threshold of 0.05/2 (number of sexual factors tested) was used to define statistical significance.

**Local genetic correlation analysis.**   We further used SUPERGNOVA [22] to detect local genetic correlation under the consideration that even with the absence of global genomic correlation, there might be specific regions in the genome contributing to both traits. SUPER-GNOVA applies an algorithm that partitions the whole genome into 2,353 LD-independent regions with an average length of 1.6 centimorgans and quantifies pairwise local genetic correlation contributed by genetic variation at each genomic region. A Bonferroni-corrected *P*-value threshold of 0.05/2,353 was used to represent statistical significance.

**Cell-type-specific enrichment analysis.**   To gain an overall impression of the relationship among sexual factors and UL, we partitioned the SNP-heritability of each trait using stratified-LDSC [23] based on cell-type-specific annotations and detected their clustering patterns. A total of 396 cell-type-specific annotations covering six histone marks (DNase, H3K27ac, H3K36me3, H3K4me1, H3K4me3, and H3K9ac) generated by the Roadmap Epigenomics project were used [24], which were further divided into 9 broad groups, namely adipose, central nervous system, digestive system, cardiovascular, musculoskeletal and connective tissue, immune and blood, liver, pancreas, and other. Enrichment values were transformed into color scale and visualized by hierarchical clustering. A Bonferroni-corrected *P*-threshold ($P < 0.05/396$) was used to define statistical significance.

**Cross-trait meta-analysis.**   Shared genetic components could be consistent with either genetic variants having independent effect on both traits (i.e., "pleiotropy") or genetic variants influencing one trait via its effect on the other (i.e., "causality") [20]. We next performed a cross-trait meta-analysis using Cross-Phenotype Association (CPASSOC) [25] to identify pleiotropic loci affecting both traits. CPASSOC offers two test statistics, $S_{Hom}$ and $S_{Het}$. While $S_{Hom}$ is efficient for homogenous genetic effects, $S_{Het}$ is an extension of $S_{Hom}$ aiming to account for heterogeneity among genetic effects which is more common in practice. Therefore, $S_{Het}$ was used in the current study.

We applied PLINK's "clumping" function [26] to obtain independent loci (parameters:—clump-p1 5e-8—clump-p2 1e-5—clump-r2 0.2—clump-kb 500). Within each locus, the variant with the lowest *P*-value was kept as the index SNP. Significant pleiotropic SNPs were defined as index variants satisfying both $P_{CPASSOC} < 5 \times 10^{-8}$ and $P_{single-trait} < 1 \times 10^{-3}$ (for both traits). Novel pleiotropic SNPs were defined as significant pleiotropic SNPs that did not reach genome-wide significance in single-trait GWASs ($5 \times 10^{-8} < P_{single-trait} < 1 \times 10^{-3}$), were independent ($r^2 < 0.20$) of previously reported genome-wide significant SNPs (of sexual factors and UL), and had no neighboring SNPs (± 500 kb) reaching $P < 5 \times 10^{-8}$ in single-trait GWASs.

We used Ensembl Variant Effect Predictor (VEP) [27] and 3DSNP [28] for detailed functional annotation of the identified pleiotropic SNPs.

**Two-sample Mendelian randomization analysis.**   We performed a bidirectional two-sample Mendelian randomization (MR) analysis to evaluate causal relationships between sexual factors and UL. The random- or fixed-effect inverse-variance weighted (IVW) method was applied as our primary approach [29]. This method pools the estimate from each IV and provides an overall estimate of the causal effect assuming all IVs to be valid; or are invalid in such a way that the overall pleiotropy is balanced to be zero. Valid IVs must satisfy three core MR assumptions [30]: (1) the IV is associated with the exposure of interest, (2) there are no confounders of the association between IV and outcome, and (3) the IV affects the outcome only through the effect on the exposure. To assess plausibility of the first assumption, we measured

the strength of each instrument by calculating the *F*-statistic using the formula, $F = \left(\frac{N-K-1}{K}\right)\left(\frac{R^2}{1-R^2}\right)$, where $R^2$ is the proportion of the variability of the trait explained by each instrument, *N* is the GWAS sample size, and *k* is the number of instruments. IVs with *F*-statistic below 10 denote the presence of weak instrument bias [31], and were therefore removed from further analyses. Additional sensitivity analyses were conducted to evaluate the second and third MR assumptions, including: (i) MR-Egger regression [32] to detect and correct for bias due to directional pleiotropy; (ii) weighted median approach [33] which provides a consistent estimate of the causal effect even when up to 50% of genetic variants are invalid; (iii) exclusion of palindromic IVs with strand ambiguity; (iv) an assessment of heterogeneity using Cochran's *Q*-statistic, where a *P*-value < 0.05 indicates the presence of heterogeneity and thus a random-effect IVW approach was used; and (v) exclusion of index SNPs with pleiotropic associations (*P*-value < 5×10$^{-8}$) with potential confounding traits (**S1, S2, and S3 Tables**) as confirmed by GWAS Catalog [34], and index SNPs identified by the cross-trait meta-analysis to be shared between sexual factors and UL.

For causal estimates that showed directional consistency across primary approach and sensitivity analyses, we further applied an IVW-based multivariable MR (MVMR) approach to adjust for any potential confounding effects (which might have masked or inflated the true causal associations) acting through AAM [35], age at natural menopause (ANM) [36], and age at first birth (AFB) [18], factors that are known to affect the lifelong exposure of a woman to steroid sex hormones. Following MVMR, a mediation analysis within a two-step MR framework was performed to explore the potential mediating effects of various hormone-related phenotypes in the sexual factors-UL relationships (**S1 Fig**). Detailed methods can be found in **S1 Text**. Given the prevalence of approximately 4 percent of ultrasound-identified UL occurs in females aged 20 to 30 years [37], there is a possibility that UL may affect subsequent sexual behaviors. We therefore conducted a reverse-direction MR to evaluate the putative causal effect of genetic liability to UL on sexual factors using UL-associated independent SNPs as IVs.

Two-sample MR analyses were conducted using packages "TwoSampleMR" (v0.5.4) and "MendelianRandomization" (v0.7.0) in software R (v4.1.0).

**Supplementary MR analysis.** To assess the robustness of findings from the two-sample MR, particularly in relation to different menopausal status, we undertook supplementary one-sample MR analysis utilizing participant-level data obtained from the UK Biobank.

In brief, UK Biobank is a large-scale prospective study that recruited 502,536 participants aged 37–73 years between 2006 and 2010 [38]. To align with our primary genetic analyses, we focused exclusively on a subset of 257,414 females of European descent. During the initial visit, participants completed touch-screen questionnaires to provide information on sexual factors, and blood samples were collected for genotyping purposes. Detailed information regarding genotyping, imputation, and quality control has been described previously [38]. For our analysis, a total of 180,494 females who had complete data on sexual factors and imputed genotypes were included. Among these participants, 14,542 were identified as UL cases based on a combination of hospital-linked medical records (ICD-10 code D25 and ICD-9 code 218) and self-report (interview with research nurse).

The 113 and 117 significantly associated SNPs were utilized to construct genetic risk score (GRS) for AFS and NSP, respectively. To mitigate one-sample bias towards the confounded observational association, we summarized the number of AFS/NSP-increasing alleles to generate the unweighted GRS rather than weighted GRS. A two-stage method was implemented to estimate the causal effect of AFS or NSP on UL risk. The first stage model consisted of a linear regression of AFS/NSP on the allele score and the second stage model consisted of a logistic regression of UL status on the fitted values from the first stage regression, with adjustment for

age at recruitment, assessment center, top ten genetic principal components, educational attainment, Townsend deprivation index, AAM, AFB, and history of oral contraceptive pill use or hormone-replacement therapy in both stages. Taking advantage of the participant-level data, we were able to perform separate analyses for pre-menopausal (n = 43,741) and post-menopausal females (n = 108,330). The categorization was based on a single question asked during the baseline assessment: "Have you had your menopause (periods stopped)?". Participants could choose from five responses, including "Yes", "No", "Not sure–had a hysterectomy", "Not sure–other reason", and "Prefer not to answer". We classified females who responded "Yes" as post-menopausal, those who responded "No" as pre-menopausal, and considered other responses as missing values, leading to their exclusion from the analysis. In the subgroup analysis of post-menopausal females, ANM was additionally adjusted.

A Bonferroni-corrected $P$-value threshold of 0.025 was employed for determining statistical significance in all MR analyses, considering the two sexual factors that were examined.

## Results

### Global and local genetic correlation

Leveraging genome-wide SNP information, we identified a significant negative genomic correlation between AFS and UL ($r_g$ = –0.11, $P$ = 7.83×10$^{-4}$). No genomic correlation was observed for NSP with UL ($r_g$ = 0.01, $P$ = 0.62).

Breaking down the whole genome into 2,353 regions, four genomic regions were identified as contributing significant local genetic correlations to AFS and UL, including 3q27.2–27.3 (chr3: 185263467–186780913), 5q35.2–35.3 (chr5: 176129392–177991305), 4q24 (chr4: 103388441–104802530), and 5q14.1–14.2 (chr5: 80481996–81712715) (**Fig 2** and **S5 Table**). Of note, local signals at 5q35.2–35.3 were previously implicated in UL, and 5q14.1–14.2 was previously implicated in AFS (assessed in GWAS Catalog on September 21, 2022). While 3q27.2–27.3 and 4q24 have not been linked to AFS or UL directly, the former region was reported to associate with AAM, the latter was reported to associate with testosterone measurement. Among the four local signals, 5q35.2–35.3 was further identified as shared between NSP and UL (**Fig 2** and **S6 Table**), despite the negligible global genomic correlation.

### Cell-type-specific heritability enrichment

In addition to region-specific genetic association, we investigated partitioned SNP-heritability enrichment using 396 cell-type-specific annotations. Three specific annotations were identified for UL with Bonferroni-corrected significant enrichment, including fetal heart in H3K9ac and DNase, and rectal smooth muscle in H3K4me1. As for AFS and NSP, they exhibited significant heritability enrichments in 29 and 17 annotations, respectively, all of which were related to the central nervous system (**S7 Table**). Notably, despite non-significance, we observed a close clustering of UL with both AFS and NSP in immune and blood related components, especially in fetal thymus and primary T cells from cord blood (**S2 Fig**).

### Cross-trait meta-analysis and pleiotropic loci

Motivated by the significant genetic overlap observed in our global and local heritability analyses, we next sought to detect pleiotropic loci at the variant level. In total, 15 independent loci reached genome-wide significance in CPASSOC ($P_{CPASSOC} < 5×10^{-8}$ and $P_{UL} < 1×10^{-3}$ and $P_{AFS/NSP} < 1×10^{-3}$), including 10 loci shared between AFS and UL, and 5 loci shared between NSP and UL (**Table 1**). Among these pleiotropic loci, SNP rs11031005 was the strongest shared signal ($P_{CPASSOC}$ = 2.20×10$^{-17}$) identified for AFS and UL, interacting with *FSHB*

A) AFS (Age at first sexual intercourse) and UL

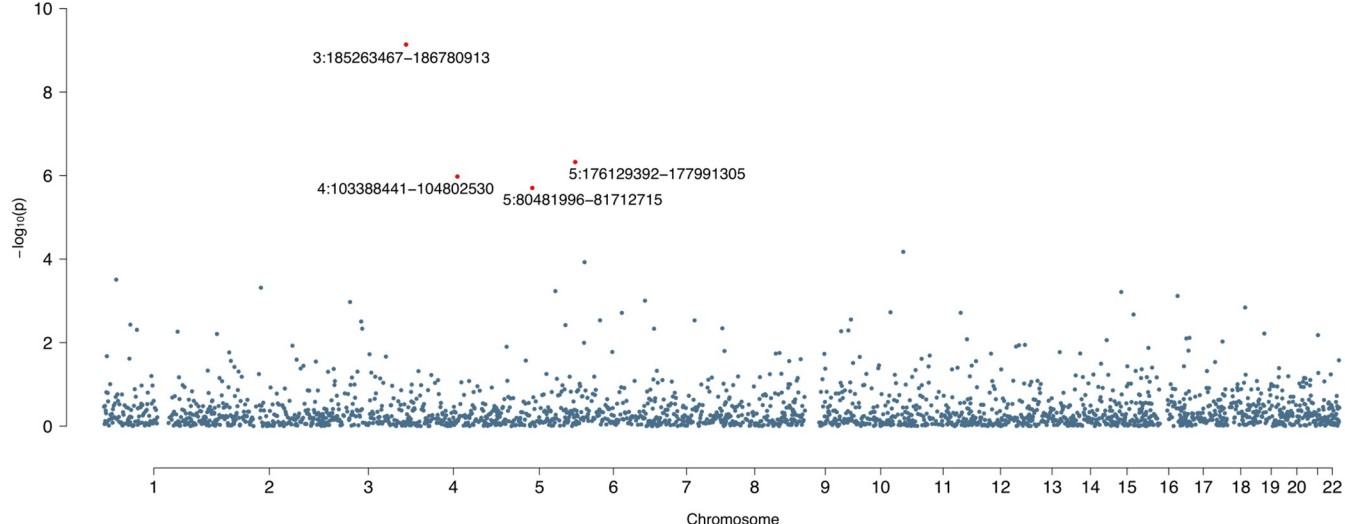

B) NSP (Lifetime number of sexual partners) and UL

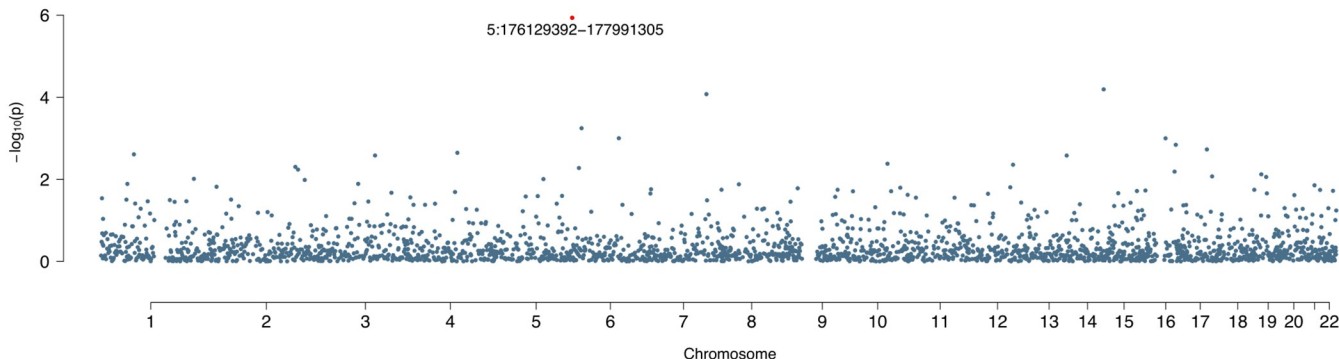

**Fig 2. Local genetic correlations between uterine leiomyoma and sexual factors. (A).** Manhattan plot presenting region-specific *P*-values for local genetic correlation between uterine leiomyoma and age at first sexual intercourse. **(B).** Manhattan plot presenting region-specific *P*-values for local genetic correlation between uterine leiomyoma and lifetime number of sexual partners. Red dots represent loci showing significant local genetic correlation after multiple testing adjustment ($P < 0.05/2,353$). AFS, age at first sexual intercourse; UL, uterine leiomyoma; NSP, lifetime number of sexual partners.

through three-dimensional chromatin looping [28]. SNP rs2018458 was the strongest shared signal ($P_{\text{CPASSOC}} < 1.00 \times 10^{-324}$) identified for NSP and UL, located near *TNRC6B* which is a well-known UL risk gene [16].

After excluding SNPs that were in LD ($r^2 \geq 0.20$) with previously reported single-trait-associated significant SNPs, we identified 4 novel SNPs (rs6669189, rs10173592, rs73250917, and rs67796606) shared between AFS and UL. While SNP rs10173592 was located in an intergenic region, it interacts with *FZD7* [28], a previously reported risk gene for AFS [18]. SNP rs6669189 was located near *TNNI3K* previously implicated in AAM [39–41] and AFS [18]; SNP rs73250917 was located near *SCFD2* previously implicated in UL [16]; and SNP rs67796606 was harbored in *ZBTB46* previously implicated in AAM [41] and AFS [18].

In addition to rs6604866 (located at the same locus with rs6669189), two novel SNPs (i.e., rs61830447 and rs5742915) were further identified as shared between NSP and UL. SNP rs61830447 was mapped to *CELF2*, a gene previously reported as associated with AFS [18]; and SNP rs5742915 was located near *PML*, a gene previously reported as associated with AAM [41].

**Table 1. Pleiotropic loci identified in cross-trait meta-analysis between sexual factors and uterine leiomyoma ($P_{CPASSOC} < 5{\times}10^{-8}$, single trait $P$-value $<1{\times}10^{-3}$).**

| Index SNP | A1/A2 | BETA | | $P$ | | $P_{CPASSOC}$ | Genomic coordinates | Linear closest genes[a] | Interacting genes[b] |
|---|---|---|---|---|---|---|---|---|---|
| | | Sexual factor | UL | Sexual factor | UL | | | | |
| *Age at first sexual intercourse and uterine leiomyoma* | | | | | | | | | |
| rs6669189 | C/T | 0.01 | 0.03 | $4.90{\times}10^{-06}$ | $3.32{\times}10^{-04}$ | $3.22{\times}10^{-08}$ | chr1:74977277–75014538 | *TNNI3K, FPGT-TNNI3K* | *C1orf173* |
| rs10173592 | T/C | -0.01 | 0.04 | $1.00{\times}10^{-05}$ | $3.01{\times}10^{-05}$ | $2.15{\times}10^{-09}$ | chr2:202863476–202952096 | - | *FZD7, LOC100652824* |
| rs6800916 | T/A | 0.02 | 0.14 | $2.50{\times}10^{-05}$ | $1.49{\times}10^{-05}$ | $2.75{\times}10^{-09}$ | chr3:49970310–50435832 | *RBM6* | *MON1A, MST1R, RBM5, RBM6* |
| rs73250917 | C/T | -0.01 | 0.04 | $2.60{\times}10^{-05}$ | $5.45{\times}10^{-05}$ | $2.27{\times}10^{-08}$ | chr4:53823221–53872674 | *SCFD2, RP11-752D24.2* | *RASL11B, DANCR, LOC152578, MIR4449, SNORA26* |
| rs11031005 | T/C | -0.01 | 0.10 | $4.20{\times}10^{-04}$ | $1.39{\times}10^{-14}$ | $2.20{\times}10^{-17}$ | chr11:30125619–30395871 | - | *FSHB* |
| rs2732538 | T/A | 0.02 | 0.05 | $4.40{\times}10^{-05}$ | $1.57{\times}10^{-05}$ | $4.60{\times}10^{-09}$ | chr11:35073408–35095948 | - | *CD44, MIR1343* |
| rs28583837 | G/A | -0.01 | -0.06 | $2.60{\times}10^{-04}$ | $2.93{\times}10^{-08}$ | $1.31{\times}10^{-11}$ | chr12:123450765–123913433 | *KMT5A* | *SETD8, CDK2AP1, RILPL2, SBNO1, SNRNP35* |
| rs9603689 | A/G | 0.01 | 0.04 | $8.10{\times}10^{-04}$ | $5.95{\times}10^{-06}$ | $2.72{\times}10^{-08}$ | chr13:40724728–40724728 | - | *LINC00332, LINC00548* |
| rs76513770 | T/C | -0.02 | 0.04 | $8.50{\times}10^{-09}$ | $8.35{\times}10^{-04}$ | $1.82{\times}10^{-09}$ | chr16:72006370–72822296 | *AC004158.2* | - |
| rs67796606 | A/G | -0.02 | -0.07 | $6.00{\times}10^{-06}$ | $3.88{\times}10^{-05}$ | $4.55{\times}10^{-09}$ | chr20:62294045–62488635 | *ZBTB46* | *SLC2A4RG, ABHD16B, ARFRP1, DNAJC5, LIME1, MIR1914, MIR647, MIR941-1, MIR941-2, MIR941-3, MIR941-4, TNFRSF6B, TPD52L2, UCKL1, UCKL1-AS1, ZBTB46, ZGPAT* |
| *Lifetime number of sexual partners and uterine leiomyoma* | | | | | | | | | |
| rs6604866 | C/G | -0.01 | -0.03 | $3.20{\times}10^{-06}$ | $2.28{\times}10^{-04}$ | $1.53{\times}10^{-08}$ | chr1:74977277–75014538 | *TNNI3K, FPGT-TNNI3K* | *C1orf173* |
| rs61830447 | A/G | 0.01 | 0.04 | $2.73{\times}10^{-05}$ | $6.43{\times}10^{-05}$ | $3.65{\times}10^{-08}$ | chr10:11134855–11166455 | *CELF2* | *CELF2-AS2, CELF2* |
| rs5742915 | T/C | -0.01 | -0.04 | $1.04{\times}10^{-04}$ | $2.02{\times}10^{-06}$ | $9.57{\times}10^{-09}$ | chr15:74336633–74353561 | *PML* | *LOXL1-AS1, GOLGA6A, ISLR, ISLR2, LOC283731, LOXL1, PML, STOML1, STRA6* |
| rs66998222 | A/G | 0.01 | -0.06 | $2.38{\times}10^{-05}$ | $4.94{\times}10^{-08}$ | $8.34{\times}10^{-11}$ | chr16:51420501–51488129 | - | *SALL1* |
| rs2018458 | A/C | 0.01 | -0.05 | $<1.00{\times}10^{-324}$ | $2.28{\times}10^{-05}$ | $<1.00{\times}10^{-324}$ | chr22:40429365–40539379 | *TNRC6B* | *LOC100130899, FAM83F, TNRC6B* |

Position is under build 37 (hg19). [a] Linear closest genes of index SNPs were mapped by using VEP. [b] 3D interacting genes of index SNPs were mapped by using 3DSNP. SNP, single nucleotide polymorphisms; UL, uterine leiomyoma.

Detailed annotations of each index SNP are shown in **S8 Table**.

## Mendelian randomization and causal relationship

Given the presence of between-IV heterogeneity detected by the Cochran's Q test (AFS to UL: $P_{Cochran's\,Q} = 1.27{\times}10^{-4}$; NSP to UL: $P_{Cochran's\,Q} = 9.27{\times}10^{-4}$), we used the random-effect IVW as our primary approach. We identified an inverse but non-significant association between genetically predicted AFS and risk of UL (OR = 0.89, 95% CI = 0.79–1.01, $P$ = 0.06), which remained directionally consistent across the complementary two-sample MR approaches (MR-Egger: OR = 0.99, 95% CI = 0.59–1.65, $P$ = 0.96; weighted median: OR = 0.95, 95% CI = 0.82–1.11, $P$ = 0.47; palindromic SNPs excluded: OR = 0.89; 95% CI = 0.78–1.01,

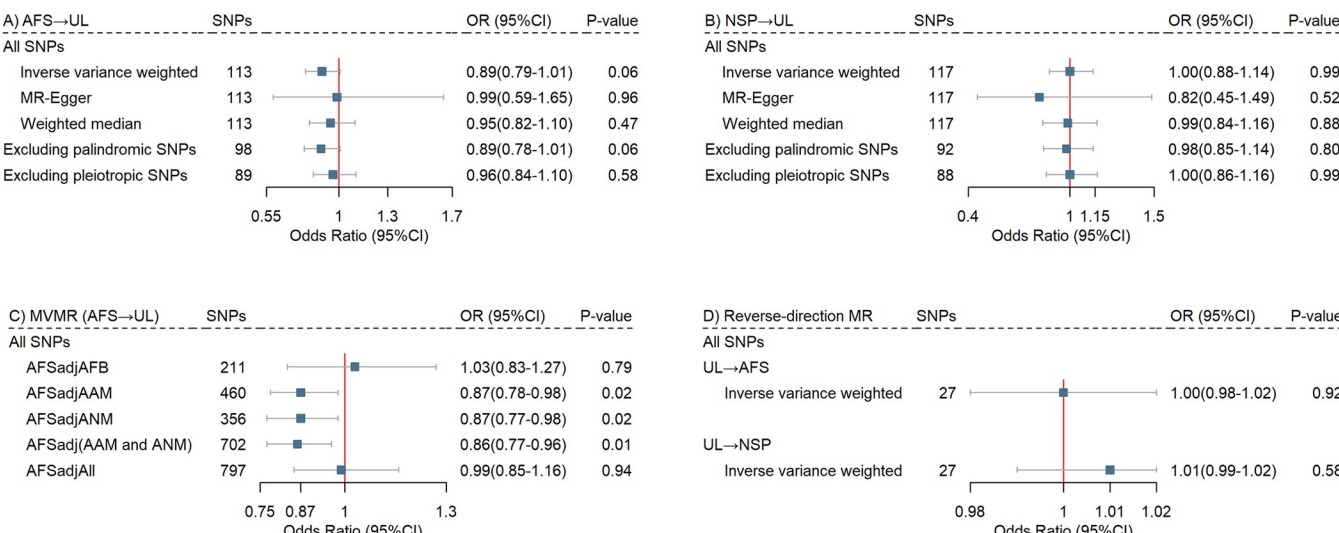

**Fig 3. Causal relationships underlying uterine leiomyoma, age at first sexual intercourse, and lifetime number of sexual partners.** **(A).** Estimates of causal effects for genetically predicted age at first sexual intercourse with uterine leiomyoma. **(B).** Estimates of causal effects for genetically predicted lifetime number of sexual partners with uterine leiomyoma. **(C).** Independent effects of genetically predicted age at first sexual intercourse on uterine leiomyoma after adjusting for each potential confounder separately and together using multi-variable Mendelian randomization. **(D).** Estimates of causal effects for genetically predicted uterine leiomyoma with sexual factors (age at first sexual intercourse, lifetime number of sexual partners). Boxes represent the point estimates of causal effects, and error bars represent 95% confidence intervals. AFS, age at first sexual intercourse; UL, uterine leiomyoma; NSP, lifetime number of sexual partners; AFB, age at first birth; AAM, age at menarche; ANM, age at natural menopause.

$P = 0.06$; pleiotropic SNPs excluded: OR = 0.96, 95% CI = 0.84–1.10, $P = 0.58$). No sign of association was observed for genetically predicted NSP with UL risk (**Fig 3**). We found little evidence of weak instrument bias ($F$-statistics for AFS 33.3; for NSP 41.0; for UL 883.5).

We further performed MVMR to evaluate the independent effect of AFS while accounting for essential confounders including AAM, ANM, and AFB. As a result, genetically predicted higher AFS was significantly associated with a lower risk of UL after adjusting for AAM (OR = 0.87, 95% CI = 0.78–0.98, $P = 0.02$), ANM (OR = 0.87, 95% CI = 0.77–0.98, $P = 0.02$), or AAM and ANM together (OR = 0.86, 95% CI = 0.77–0.96, $P = 0.01$). Nonetheless, when controlling for the effect of AFB, the AFS-UL association attenuated to null (OR = 1.03, 95% CI = 0.83–1.27, $P = 0.79$). Similar null result was generated after adjusting for AAM, ANM, and AFB in combination (OR = 0.99, 95% CI = 0.85–1.16, $P = 0.94$) (**Fig 3**). Mediation analysis did not support any hormone-related phenotypes as mediators in the relationship between AFS/NSP and UL. However, significant indirect effects of AFS on UL were observed through several hormone-related phenotypes, including AFB (**S9 Table**).

No evidence of reverse causality was found such that genetic liability to UL neither influences AFS (OR = 1.00, 95% CI = 0.98–1.02, $P = 0.92$) nor NSP (OR = 1.01, 95% CI = 0.99–1.02, $P = 0.58$) (**Fig 3**).

Corroborating the two-sample MR findings, GRS-based one-sample MR analysis demonstrated no evidence supporting a causal effect of genetically predicted AFS or NSP on risk of UL, consistent among both pre-menopausal and post-menopausal females (OR = 1.00 and $P > 0.10$ across all analyses).

## Discussion

To the best of our knowledge, this is the first large-scale genome-wide cross-trait analysis which comprehensively interrogated the shared genetic architecture between sexual factors

and UL. From a genetic perspective, our work demonstrates a biological link between AFS, NSP, and UL that is driven by pleiotropic effects rather than causal relationships.

Overall, a limited yet statistically significant global genetic similarity was identified between AFS and UL, where ~ 11% of the genetic contribution to AFS is shared with the genetic contribution to UL. The overall genetic correlation was further corroborated by significant local signals identified at multiple specific genomic regions, collectively highlighting a non-trivial shared biological basis. Notably, although we identified the global $r_g$ between NSP and UL to be close to zero, subsequent regional analyses revealed a statistically significant, shared local heritability in a genomic region that was previously reported to be independently associated with UL, AAM, and ANM by separate GWASs [16,17,42–44]. As such, our results show that global genomic approaches to measuring shared heritability may fail to detect meaningful overlap between association signals locally.

Through analyzing gene expression data with GWAS summary statistics, our study confirmed previous identifications of a brain cell-type enrichment for both AFS and NSP [18,19]. The similar enrichment patterns when partitioning heritability for UL and that of sexual factors further elucidate potential overlapped etiology related to the immune system. A chronically active inflammatory immune system is suggested to be involved in UL formation [45,46], and sexual factors are linked to sex hormones [6,7] which have well-established impacts on immune responses [47]. Future work is warranted to fully uncover specific pathways that underlie the UL-AFS/NSP biological associations.

A non-negligible shared genetic basis, as reflected by the significant genetic correlations and clustering of cell-type-specific annotations, can either be due to pleiotropic effects (horizontal pleiotropy), causal relationship (vertical pleiotropy), or both. The 10 pleiotropic loci identified for AFS with UL as well as the 5 pleiotropic loci identified for NSP with UL reflected horizontal pleiotropy. However, it is worth noting that there is a lack of overlap between the identified pleiotropic loci and shared regions, which could be attributed to differences in the focus, scale of analysis, and the power to detect significant associations. One advantage of meta-analyzing GWASs of different traits is that combining association evidence across multiple studies can reveal signals which might not have reached genome-wide significance in a single-trait analysis. Indeed, we identified 7 novel loci to jointly affect AFS/NSP and UL, among which we highlight three interesting examples, *FZD7*, *CELF2*, and *TNNI3K*.

Harboring the most significant novel SNP shared between AFS and UL, *FZD7* encodes a seven-pass transmembrane wnt (Wingless/Integrated) receptor which plays an essential role in the development of various tumors [48]. However, the involvement of *FZD7* in UL has rarely been studied. By interacting with *SIRT1*, a gene that regulates steroid hormone receptor activity, *FZD7* may also participate in estrogen receptor signaling [49]. Encoding a nucleocytoplasmic shuttling RNA-binding protein, *CELF2* is involved in development of the central nervous system, notably, the disruption of cortical development related to decision-making [50]. *CELF2* also functions in tumor initiation and progression given its consistently reduced expression during neoplastic transformation [51]. In addition, expression of *CELF2* is increased in response to T-cell signaling, suggesting its involvement in inflammatory and immune responses [52]. As a novel shared locus identified for both AFS-UL and NSP-UL, *TNNI3K* encodes a troponin I interacting kinase mainly implicated in cardiac physiology (consistent with our findings that UL is highly enriched in heart and muscle-relevant cell types) [53]. Several as of yet not replicated studies on carcinogenesis [54], viral infection [55], and obesity [56] further expand the spectrum of potential roles of *TNNI3K*. Future experimental work is needed to provide more detailed functional annotation of the newly discovered loci as they related to sexual factors and UL.

Contrary to the evidence observed for horizontal pleiotropy, we found no evidence of a causal relationship among these traits investigated. Interestingly, when accounting for the effect of AAM or ANM, a significant negative association of genetically predicted AFS on UL was shown, indicating a later age at first sex being a putatively protective factor of UL initiation. However, our MVMR exploration results suggest that this association is predominantly confounded or mediated by the effect of AFB, which is further supported by the results of a supplementary mediation analysis (an interpretation and discussion of the mediation results can be found in **S1 Text**). Both AFS and AFB mark the onset of human reproductive behavior and have important implications for reproductive health. While AFB has long been associated with UL risk by epidemiological studies, for example, the inverse associations reported by the longitudinal Nurses' Health Study II (AFB $\geq$ 25 years compared with $\leq$ 24 years: 10%-30% risk reduction; AFB $\geq$ 31 years: RR = 0.71, 95% CI = 0.64–0.79, [57,58], evidence on AFS remains scarce.

Leveraging genetic information, our work demonstrates that AFS alone is unlikely to pose a direct causal effect on UL initiation, which is concordant with findings by Pakiz *et al.* that AFS was not an independent predictor of UL [15]. While infectious agent exposure (closely related to AFS/NSP) represents an important source of inflammatory reactions, myometrial injury, and tissue repair or regeneration believed to impact UL pathogenesis [46,59], based on current results we question whether intrauterine infection may contribute to the occurrence of UL, which has been a topic of debate and focus of research in the UL field [10,12–14, 60–63]. Nonetheless, although we have used genetic associations as proxies for AFS and NSP, direct investigations on reproductive tract infections or pelvic inflammation are limited which may have precluded interpretation of our findings.

The key strengths of this study include the largest combination of available datasets, the employment of multiple analytic frameworks to investigate the genetic-based association, and the use of both two-sample and one-sample MR designs to assess the causal effects of sexual factors on UL risk. However, we also acknowledge potential limitations. First, as our results were restricted to samples of European ancestry, generalizability to other ancestry groups is precluded. Studies leveraging large-scale GWAS that encompass different ancestries are required to understand better the genetic associations across diverse populations. Second, although we have carefully selected genetic instruments that show strong statistical associations with the exposures, common SNPs can only account for a small proportion of the overall variability observed in complex traits. This limitation is particularly evident in the case of behavioral factors such as AFS and NSP, which are influenced by a multitude of factors, including environmental, demographic, and genetic factors, as well as their intricate interactions [19]. To obtain a more comprehensive understanding of how genetics and the environment jointly shape sexual behaviors and their potential impact on UL, future research integrating genetic analyses with robust environmental assessments is needed. Third, we conducted our primary analyses using sex-combined GWAS summary data of NSP, which may introduce sex heterogeneity. However, the high magnitude of genetic correlation ($r_g$ = 0.82) between the sex-specific GWASs of the general risk tolerance measure [19] has largely justified the approach of pooling males and females to maximize statistical power; furthermore, global genetic correlation and two-sample MR analyses using female-specific summary statistics for NSP (non-formal publication; http://www.nealelab.is/uk-biobank/) yielded largely consistent results ($r_g$ = 0.03, $P$ = 0.50; NSP to UL: $OR_{IVW}$ = 0.88, $P$ = 0.50; UL to NSP: $OR_{IVW}$ = 1.01, $P$ = 0.50), as in our one-sample MR analysis restricting to UK Biobank female participants. Future investigations leveraging sex-specific summary-level data are needed to validate our findings. Fourth, a considerable proportion of sample overlap may exist among the GWAS datasets used in the current study. However, the genetic approaches (i.e., LDSC,

SUPERGNOVA, CPASSOC) we applied are not subject to bias introduced by sample overlap, and any bias caused by sample overlap in two-sample MR may falsely lead to a significant association, which was not observed in our analysis. Future validation using independent datasets is warranted. Moreover, we acknowledge that our study's definition of AFS and NSP, based on the UK Biobank data, includes a broad range of sexual experiences. This lack of specificity regarding the type and context of sexual activity may limit our understanding of the nuanced variations in risk associated with different sexual behaviors. Future research should explore more specifically defined AFS and NSP to assess their impact on the risk of UL and other relevant outcomes. It is also important to acknowledge the sensitive nature of our work and to emphasize that our intent is to contribute to a larger body of literature aimed at promoting positive health outcomes for women. Stigmatization or judgment of individual sexual behavior has no place in research. To optimize statistical power, our study, as well as the previous UL GWAS [16], incorporated UL cases that were ascertained through self-report, introducing the possibility of recall bias in outcome definition. At last, we did not investigate UL subtypes based on molecular characteristics or clinical factors (e.g., location, multiplicity, and size) due to limited well-powered GWAS available for these traits. Given that previous research has identified various associations of AFS with singular or multiple UL occurences [15,64], future subtype-specific GWASs are needed to extend our findings.

## Conclusions

To conclude, leveraging large-scale genetic variation data for each trait, our study demonstrates a biological link underlying sexual factors (AFS and NSP) and UL. From a genetic perspective, sexual factors are not likely to elevate the risk of UL, but rather overlapping biological mechanisms underlie the relationship between these phenotypes. Future studies are needed to confirm our findings, and to elucidate the specific mechanisms involved in the shared genetic influences and their potential impact on UL development.

## Supporting information

**S1 Text. Supplementary methods for mediation analysis and an interpretation and discussion of the mediation analysis results.**
(DOCX)

**S1 Fig. Graphical representation of proposed mediation through hormone-related phenotypes in the association of sexual factors with uterine leiomyoma.**
(DOCX)

**S2 Fig. Clustering of cell-type-specific annotation for uterine leiomyoma, age at first sexual intercourse, and lifetime number of sexual partners over histone marks.**
(DOCX)

**S1 Table. Characteristics of genetic instruments identified by GWAS to be associated with age at first sexual intercourse.**
(XLSX)

**S2 Table. Characteristics of genetic instruments identified by GWAS to be associated with lifetime number of sexual partners.**
(XLSX)

**S3 Table. Characteristics of genetic instruments identified by GWAS to be associated with uterine leiomyoma.**
(XLSX)

**S4 Table. Data sources, sample sizes, number of IVs, and F-statistics.**
(XLSX)

**S5 Table. Results from local genetic correlation analysis conducted for age at first sexual intercourse and uterine leiomyoma.**
(XLSX)

**S6 Table. Results from local genetic correlation analysis conducted for lifetime number of sexual partners and uterine leiomyoma.**
(XLSX)

**S7 Table. Cell-type-specific annotations showing Bonferroni-corrected significant heritability enrichments for uterine leiomyoma, age at first sexual intercourse, and lifetime number of sexual partners.**
(XLSX)

**S8 Table. Detailed annotation of genome-wide significant SNPs identified from cross-trait meta-analysis.**
(XLSX)

**S9 Table. Results of mediation Mendelian randomization analysis on sexual factors and risk of uterine leiomyoma using the "product of coefficients" method.**
(XLSX)

## Author Contributions

**Conceptualization:** Xueyao Wu, Changfeng Xiao, Xia Jiang.

**Formal analysis:** Xueyao Wu, Changfeng Xiao.

**Funding acquisition:** Cynthia Casson Morton.

**Supervision:** Xia Jiang, C. Scott Gallagher.

**Visualization:** Xueyao Wu, Changfeng Xiao.

**Writing – original draft:** Xueyao Wu, Xia Jiang.

**Writing – review & editing:** Xueyao Wu, Changfeng Xiao, Danielle Rasooly, Xunying Zhao, Cynthia Casson Morton, Xia Jiang, C. Scott Gallagher.

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
