## [Decision Letter · Decision Letter 0]

29 Nov 2023

Dear Dr Wu,

Thank you very much for submitting your Research Article entitled 'A comprehensive genome-wide cross-trait analysis of sexual factors and uterine leiomyoma' to PLOS Genetics.

The manuscript was fully evaluated at the editorial level and by independent peer reviewers. The reviewers appreciated the attention to an important problem, but raised some substantial concerns about the current manuscript. Based on the reviews, we will not be able to accept this version of the manuscript, but we would be willing to review a much-revised version. We cannot, of course, promise publication at that time. One major concern that requires attention is the critique by reviewer 2 that questions the use of a genetic instrument to study this problem. 

If you decide to revise the manuscript for further consideration at PLOS Genetics, please aim to resubmit within the next 60 days, unless it will take extra time to address the concerns of the reviewers, in which case we would appreciate an expected resubmission date by email to plosgenetics@plos.org.

We are sorry that we cannot be more positive about your manuscript at this stage. Please do not hesitate to contact us if you have any concerns or questions.

Yours sincerely,

Digna Velez Edwards

Guest Editor

PLOS Genetics

Scott Williams

Section Editor

PLOS Genetics

Please address the concerns regarding the use of genetics for traits determined by environment for this type of study and methodologic concerns.

Reviewer's Responses to Questions

**Comments to the Authors:**

Reviewer #1: This is a well-written manuscript that explores genome-wide and local genetic correlations between uterine leiomyoma (UL) and two sexual factors: age at first sexual intercourse (AFS) and lifetime number of sexual partners (NSP). The study was carried out using GWAS summary data from European-ancestry cohorts as well as individual-data from the UK Biobank to conduct Mendelian randomization (MR).

I have the following questions/comments:

1.) A significant negative genome-wide correlation was found between AFS and UL. In addition, four significant local genetic correlations were identified between these two traits. Do these local correlations have the same direction (i.e., negative) as the genome-wide correlation?

2.) Could the authors speculate a little bit in the discussion about the shared immune and blood-related components between UL with AFS and NSP?

3) Please specify if the pleiotropic AFS/UL SNPs in Table 1 are in the same genomic regions with significant local genetic correlations between AFS and UL. If that is not the case, please speculate why.

Reviewer #2: The review is uploaded as an attachment.

Reviewer #3: Thank you for this thorough analysis and exploration of sexual factors and UL. I appreciated the complementary approaches taken for this analysis and the summary of methods shown in Figure 1.

Please provide sample sizes for subgroups of pre- and post-menopausal women.

It is mentioned that the observed association between AFS and UL may be confounded by AFB. For future research, consider a mediation analysis to analyze the pathway AFS  AFB  UL.

Typos:

p. 4, lines 105-106; please keep the verbs in the same tense (past or present)

p. 8, lines 207; “… adjust for potential confounding effect…” should be “to adjust for any potential confounding effects…”

p. 15, line 408; “well-power” should be “well-powered”

Figure 1: The box “Pleiotropic effect and casual relaitonship identification” should be “Pleiotropic effect and causal relationship identification”

**Have all data underlying the figures and results presented in the manuscript been provided?**

Reviewer #1: Yes

Reviewer #2: Yes

Reviewer #3: Yes

PLOS authors have the option to publish the peer review history of their article (what does this mean?). If published, this will include your full peer review and any attached files.

Reviewer #1: No

Reviewer #2: No

Reviewer #3: No

---

## [Decision Letter · Decision Letter 1]

19 Mar 2024

Dear Dr Wu,

Thank you very much for submitting your Research Article entitled 'A comprehensive genome-wide cross-trait analysis of sexual factors and uterine leiomyoma' to PLOS Genetics.

The manuscript was fully evaluated at the editorial level and by independent peer reviewers. The reviewers appreciated the attention to an important topic but identified some concerns that we ask you address in a revised manuscript.

We therefore ask you to modify the manuscript according to the review recommendations. Your revisions should address the specific points made by each reviewer.

Yours sincerely,

Digna Velez Edwards

Guest Editor

PLOS Genetics

Scott Williams

Section Editor

PLOS Genetics

Reviewer's Responses to Questions

**Comments to the Authors:**

Reviewer #1: I appreciate the authors for addressing all of my comments. The manuscript has greatly improved after these changes.

Reviewer #3: Thank you for your careful attention to the suggestions made.

Please include a brief paragraph in the Methods section about the mediation analysis.

Reviewer #4: The authors have conducted a well thought out study and made significant improvements to the manuscript with the revised changes. I have some follow up questions/comments for them to consider.

1. In the one-sample MR analysis for post-menopausal females, did the authors exclude individuals who had a hysterectomy?

2. The authors state that none of the risk factors investigated in the mediation analysis showed supporting evidence as mediators, but then they also say that they observed a significant indirect effect of AFS on UL through AFB. It is unclear to me how they came to this conclusion when several of the potential mediators they analyzed had indirect effects with a p-value < 0.05. Please clarify.

3. The authors should explain what significance threshold they used for the mediation and MR analyses. They should also consider whether a Bonferroni correction should be applied due to the number of tests performed in their analysis.

4. The methods for the mediation analysis are not described. It would be helpful to understand how the hormone datasets were made since participants in the UK Biobank were recruited between ages 40 to 69. Hormone levels can change with age and the traits being studied can occur at different periods throughout an individual’s life. Do the authors have any thoughts on if this could influence the results?

5. UK Biobank data was used for both the UL and sexual factors GWAS. Having overlapping samples in the two-sample MR analysis when non-overlapping samples are required, could greatly bias the results. Although this was briefly discussed as a limitation, considering the impact of this, it should be thoroughly addressed. Did the authors explore any ways to avoid this, such as by using a leave-one-out analysis for the UL GWAS or using an independent dataset?

**Have all data underlying the figures and results presented in the manuscript been provided?**

Reviewer #1: Yes

Reviewer #3: Yes

Reviewer #4: Yes

PLOS authors have the option to publish the peer review history of their article (what does this mean?). If published, this will include your full peer review and any attached files.

Reviewer #1: No

Reviewer #3: No

Reviewer #4: No

---

## [Editor Report · Decision Letter 2]

22 Apr 2024

Dear Dr Wu,

We are pleased to inform you that your manuscript entitled "A comprehensive genome-wide cross-trait analysis of sexual factors and uterine leiomyoma" has been editorially accepted for publication in PLOS Genetics. Congratulations!

Yours sincerely,

Digna Velez Edwards

Guest Editor

PLOS Genetics

Scott Williams

Section Editor

PLOS Genetics

Comments from the reviewers (if applicable):

**Data Deposition**

http://datadryad.org/submit?journalID=pgenetics&manu=PGENETICS-D-23-01080R2

**Press Queries**

---

## [Editor Report · Acceptance letter]

29 Apr 2024

PGENETICS-D-23-01080R2 

A comprehensive genome-wide cross-trait analysis of sexual factors and uterine leiomyoma 

Dear Dr Wu, 

We are pleased to inform you that your manuscript entitled "A comprehensive genome-wide cross-trait analysis of sexual factors and uterine leiomyoma" has been formally accepted for publication in PLOS Genetics! Your manuscript is now with our production department and you will be notified of the publication date in due course.

With kind regards,

Zsofia Freund

PLOS Genetics

On behalf of:
